# Transcriptomic Analysis Reveals Key Pathways Influenced by HIV-2 Vpx

**DOI:** 10.3390/ijms26083460

**Published:** 2025-04-08

**Authors:** Zsófia Ilona Szojka, Balázs Kunkli, Irene Wanjiru Kiarie, Tamás Richárd Linkner, Aya Shamal Al-Muffti, Hala Ahmad, Szilvia Benkő, Marianne Jansson, József Tőzsér, Mohamed Mahdi

**Affiliations:** 1Laboratory of Retroviral Biochemistry, Department of Biochemistry and Molecular Biology, Faculty of Medicine, University of Debrecen, 4032 Debrecen, Hungary; zsofia.szojka@gmail.com (Z.I.S.); kiarie.irene@med.unideb.hu (I.W.K.); linkner.tamas@science.unideb.hu (T.R.L.); aya.muffti@med.unideb.hu (A.S.A.-M.); 2Division of Medical Microbiology, Department of Laboratory Medicine, Lund University, Box 188, SE-221 00 Lund, Sweden; marianne.jansson@med.lu.se; 3Doctoral School of Molecular Cell and Immune Biology, University of Debrecen, 4032 Debrecen, Hungary; hala.ahmad@med.unideb.hu; 4Laboratory of Inflammation-Physiology, Department of Physiology, Faculty of Medicine, University of Debrecen, 4032 Debrecen, Hungary; benkosz@med.unideb.hu

**Keywords:** HIV-2, Vpx, host response, viral pathogenesis, transcriptional regulation, cytokines

## Abstract

Viral protein X (Vpx) is a unique accessory protein encoded by the genome of the human immunodeficiency virus type 2 (HIV-2) and lineages of the simian immunodeficiency virus of sooty mangabeys. So far, counteracting the cellular restriction factor SAMHD1 and mediating the efficient translocation of viral pre-integration complex have been recognized as key functions of Vpx; however, a thorough exploration of its effects on the cellular transcriptome and cytokine milieu has not yet been undertaken. In this study, we carried out the transcriptomic analysis of THP-1 cells and determined differential gene expressions induced by HIV-2 Vpx, utilizing vectors coding for the wild-type and K68-R70 functionally restricted proteins. Significantly altered genes were then validated and quantified through real-time quantitative PCR (qPCR); additionally, replication-competent virions were also used to confirm the findings. Moreover, we analyzed the effect of Vpx expression on the secretion of key cytokines in the medium of transfected cells. Our findings revealed that wild-type HIV-2 Vpx can significantly alter the expression of genes coding for helicases, zinc finger proteins, chaperons, transcription factors and proteins involved in DNA methylation. Differentially altered genes were involved in negative regulation of viral processes, the type I interferon-signaling pathway, DNA-template transcription, elongation, the positive regulation of interferon beta production and the negative regulation of innate immune response. Importantly, Vpx was also found to decrease the expression of HIV-1 Tat, possibly through the downregulation of a crucial splicing factor, required for the maturation of Tat. Additionally, studies on cellular cytokine milieu showed that this accessory protein induced key proinflammatory cytokines. Our study provides important information about the complex role played by HIV-2 Vpx in priming and taming the cellular environment to allow for the establishment of the infection.

## 1. Introduction

The human immunodeficiency viruses types 1 and 2 (HIV-1 and HIV-2) share similarities in terms of their structural organization and genetic background [1], while differing in replication dynamics and clinical progression of infection [2]; additionally, the variability of the genetic sequences of the two viruses account for the presence of unique accessory genes [1]. Viral accessory proteins, such as the negative effector (Nef), viral infectivity factor (Vif), viral protein r (Vpr), viral protein u (Vpu) and viral protein x (Vpx), are thought to augment infectivity and play a crucial role in modulating the host cell’s immune response [3,4]. The HIV-2 genome encodes Vpx, which is known to play an important role in the pathogenesis of infection through the suppression of antiviral mechanisms of the host cell [5,6].

It is speculated that this accessory protein originated as a duplicate of HIV-1’s Vpr, given the similarity in amino acid sequence between the two proteins and their maturation process [7]. Both proteins are specifically packaged into viral particles through interaction with the p6 domain of the viral Gag precursor and mediate their intracellular function through association with the DDB1- and CUL4-associated factor 1 (DCAF1), damage specific DNA binding protein 1 (DDB1), cullin4 (Cul4) and Ring box protein1 containing E3 ubiquitin ligase complex (Rbx1) [8,9]. Association with the E3 ubiquitin ligase complex allows Vpx to target host proteins for proteasomal degradation, mainly the sterile alpha motif and HD domain protein 1 (SAMHD1); a dGTP-regulated deoxynucleoside triphosphohydrolase, which catalyzes the hydrolysis of available deoxyribonucleotide triphosphates (dNTPs). SAMHD1 is abundant in cells of the myeloid lineage [10]; however, previous studies have indicated reduced basal levels of SAMHD1 in monocyte-derived macrophages [5,11].

The depletion of the cellular dNTP pool would interfere with viral reverse-transcription; therefore, counteracting the actions of SAMHD1 by HIV-2 Vpx in non-dividing myeloid cells and resting CD4+ T cells has long been regarded as the main function of Vpx, potentiating viral infection [5,12,13]. This property has prompted exploratory research into the potential application of Vpx to augment lentiviral transduction in gene therapy, particularly for targeting latent HIV-1 reservoirs within non-dividing resting cells, such as hematopoietic stem cells (HSCs) and immune cells [14]. The Vpx-mediated enhancement of macrophage transduction has demonstrated significant promise in advancing cancer immunotherapy, enabling more efficient genetic modifications and improving their anti-tumor capabilities [15]. It is worth mentioning that other viruses, besides HIV, were found to counteract SAMHD1, either directly through post-translational modifications, such as phosphorylation by herpes viruses [16], or indirectly modulating its activity, such as in the case of HTLV-1 and HBV [17,18].

In THP-1 monocytic cells, SAMHD1 expression was associated with enhanced cell proliferation, a shift in cell cycle distribution, and a decrease in apoptotic events, indicating a major role SAMHD1 in the regulation of the cell cycle [19]. Research in cancer biology has shown that SAMHD1 supports DNA repair processes and protects cells from the adverse consequences of dNTP imbalances. The disruption or loss of the SAMHD1 function has been demonstrated to impair these protective and regulatory functions [20,21].

Moreover, Vpx-induced ubiquitination results in the proteosomal degradation of the human silencing hub (HUSH) complex, which compensates for transcription inhibition through the NF-κB antagonism of Vpx [22,23]. Additionally, Vpx facilitates the efficient translocation of a viral pre-integration complex (PIC) from the cytoplasm to the nucleolus, a process involving interactions between the Vpx and heat shock protein 40 (Hsp40) [24]. The Vpx-mediated nuclear transport of PIC is a complex process, mediated by the nuclear targeting sequences in both the N- and C- terminal domains of Vpx. Functional and efficient nuclear transport by Vpx has been shown to be critically dependent on physical interaction with nucleoporin 153 (Nup153) [25,26,27]. The mechanism of Vpx’s nuclear import, however, remains poorly understood.

The Vpx amino acid sequence contains two basic residues (K68 and R70), which are highly conserved in HIV-2 and SIVs [28]. The substitution of lysine with alanine at position 68 impaired Vpx’s role in the degradation of SAMHD1 [29], while the replacement of arginine with alanine in position 70 affected the stability of Vpx and impaired its ability to engage in nuclear localization.

Besides the suppression of the NF-κB pathway, Vpx was also shown to possess immunomodulatory functions, through interaction with the interferon regulatory factor 5 (IRF5), limiting its transactivation ability and inhibiting the immune response [22,30].

In our previous study, we observed that Vpx played an important role in the dampening of HIV-1’s infectivity in a dual HIV-1 and -2 pseudovirion transduction cell culture model [31]. This suppression of HIV-1’s infectivity was mediated through a non-direct inhibition of HIV-1 reverse transcriptase (RT) enzyme, possibly by Vpx, and the incorporation of this accessory protein into HIV-1 pseudovirions. We were intrigued by our previous findings and thereafter set out to explore other potential involvement(s) of this unique accessory protein in intracellular processes during infection; additionally, we also aimed to elucidate the effects of its expression on the cellular cytokine milieu in THP-1 cells.

## 2. Results

Transfection efficiency in THP-1 cells and the expression of the GFP-Vpx fusion protein were confirmed by fluorescence microscopy and flow cytometry and ranged from 70 to 80%. To explore the effect of HIV-2 Vpx expression on the transcriptomic profile of the cells, RNA-seq analysis was carried out on RNA isolated from cells transfected with the wild-type and functionally restricted mutant (K68A-R70A) Vpx expressing constructs, in conjugation with an N-terminal GFP. Additionally, analysis was also performed on cells transfected with the GFP expressing mock control. RNA-seq yielded 15,353,387 to 20,619,520 raw sequencing reads per sample, and after optimization, 8,490,166 to 11,769,737 uniquely mapped reads were obtained (Appendix A). Expressed transcripts per sample were determined with a minimum threshold of five counts per gene in order to be considered expressed (Appendix A), whereas the regulation pattern of 175 genes was only altered in the presence of wild-type Vpx (Appendix A). Reads were aligned to human reference genome GRCh38.p13 using Bowtie 2, and 61.13 to 71.24% of reads were uniquely mapped per sample.

### 2.1. Differential Gene Expression Induced by Wild-Type Vpx

The analysis of Vpx-transfected THP-1 cells revealed that compared to the mock control, Vpx significantly altered the regulation pattern of 864 genes, out of which 606 genes were upregulated and 260 were downregulated. Of the 606 upregulated genes, 576 genes were induced by the wild-type Vpx, while only 30 genes were differentially induced by the functionally restricted K68A-R70A mutant. In regard to the 260 downregulated transcripts, the wild-type Vpx resulted in the downregulation of 249 genes, compared to only 11 that were downregulated by the functionally restricted protein. Wild-type Vpx resulted in the up-regulation of protocadherin gamma subfamily C, 4 (PCDHGC4), caspase 3 (CASP3), U2 small nuclear RNA auxiliary factor 1-like 5 (U2AF1L5); NOP58 ribonucleoprotein (NOP58); sorting nexin 14 (SNX14); monocyte to macrophage differentiation associated (MMD); RANBP2-like and GRIP domain containing 2 (RGPD2); serine/threonine kinase 17a (STK17A); gamma-secretase activating protein (GSAP); and lysine-rich coiled-coil 1 (KRCC1), while the expression of the SKI family transcriptional corepressor 2 (SKOR2); CORO7-PAM16 read-through (CORO7-PAM16); Mucin 4, cell surface associated (MUC4), U2 small nuclear RNA auxiliary factor 1 (U2AF1); pleckstrin homology domain containing A1 (PLEKHA1); NALCN channel auxiliary factor 2 (NALF2); LDL receptor related protein 4 (LRP4); proline rich basic protein 1 (PROB1); serpin family A member 1 (SERPINA1); and neuronal pentraxin receptor (NPTXR) were downregulated in the presence of wild-type HIV-2 Vpx (Appendix A).

On the other hand, transfection with a construct coding for the functionally restricted K68A-R70A Vpx resulted in the upregulation of several keratin encoding genes (KRT81, KRT5, KRT19, KRT6A), mal, T cell differentiation protein 2 (MAL2), serpin family B member 5 (SERPINB5), MAGE family members A3 and A6 (MAGEA3 and MAGEA6), kallikrein-related peptidase 6 (KLK6) and the ras homolog family member D (RHOD). Several genes including eukaryotic translation initiation factor 3 subunit C-like (EIF3CL), cannabinoid receptor 2 (CNR2), the SUMO peptidase family member, NEDD8-specific (SENP8), torsin 1A interacting protein 2 (TOR1AIP2), par-6 family cell polarity regulator gamma (PARD6G), zinc finger protein 490 (ZNF490), radial spoke head 3 (RSPH3), solute carrier family 35 member E3 (SLC35E3), proline–serine–threonine phosphatase interacting protein 2 (PSTPIP2) and short stature homeobox (SHOX) were downregulated. As in the case of wild-type Vpx, the SKOR2 gene was also downregulated (Appendix A).

GO analysis identified 72 biological processes (BP) groups, 28 cellular component (CC) groups, and 27 molecular function (MF) groups among the DEGs. The top seven GO terms are shown in Appendix A. BP analysis indicated that the affected genes were mainly involved in thenegative regulation of viral process (GO:0048525), type I interferon signaling pathway (GO:0060337), spliceosomal snRNP assembly (GO:0000387), the regulation of transcription elongation from RNA polymerase II promoter (GO:0034243), the negative regulation of innate immune response (GO:0045824), the positive regulation of interferon beta production (GO:0032728) and DNA-templated transcription elongation (GO:0006354) (Appendix A). BP analysis regarding the functionally restricted K68A-R70A mutant is presented in Appendix A.

Overall, the most significantly altered pathways identified in this analysis were analyzed further, and the association between the DEGs and the top seven selected GO terms was studied. Analysis revealed that the identified DEGs were mostly involved in the regulation of transcription elongation from RNA polymerase II promoter (GO:0034243), DNA-templated transcription elongation (GO:0006354), ubiquitin ligase complex (GO:0000151), nuclear speck (GO:0016607), ubiquitin binding (GO:0043130) and the negative regulation of innate immune response (GO:0045824). The relationship between the DEGs and GO terms, as well as the logFC of the DEGs, is shown in Figure 1 (wild-type) and Appendix A (K68A and R70A mutant).

Association between 99 DEGs and the top 7 selected GO terms: (1) DNA-templated transcription elongation; (2) the negative regulation of innate immune response and (3) the regulation of transcription elongation from RNA polymerase II promoter classified under biological processes; (4) nuclear speck and (5) ubiquitin ligase complex belonging to cellular components; (6) transcription corepressor activity and (7) ubiquitin binding pertaining to molecular functions, along with the corresponding log_2_FC values of genes are represented in GOChord plot. Genes associated only with nuclear speck or ubiquitin ligase complex in the range of −0.58 < log_2_FC < 0.58 (0.58 ≈ log_2_1.5) were omitted. The color intensities of the rectangles reflect the significance of change in the corresponding gene level relative to the control (color scale: log FC = −2 blue, log FC = 0 white and log FC = 1 red).

Given the findings from GO analysis, wherein several DEGs affected by Vpx were found to play a role in the negative regulation of immune system, we also carried out analysis on the medium of transfected cells, assaying for key proinflammatory cytokines; such as Interleukin-6 (IL-6), Interleukin-1 beta (IL-1β), and transforming growth factor β (TGFβ). Our results indicate that transfection with wild-type HIV-2 Vpx resulted in an increased production of all of those cytokines, with the exception of antiviral type I interferons (IFN α), where Vpx did not alter their levels in comparison to the control (Figure 2A–D).

### 2.2. Effect of Vpx on Expression of HIV-1 Tat

To determine the effect of HIV-2 Vpx on Tat expression, we transfected HEK-293T cells with pcDNA3.1-Vpx-NeGFP, followed by transfection with an HIV-1 Tat expressing plasmid. The Western blotting of cell lysate revealed that in the presence of Vpx, the expression of Tat was decreased, compared to the Tat expressed in mock-transfected cells (Figure 3).

### 2.3. RT-qPCR Results of SKOR2, U2AF1 and CASP3 Genes

From the Gene Ontology (GO) analysis, it was observed that three genes with potential influence on the replication dynamics of HIV-2 were significantly altered. To investigate the specific effects of Vpx on the gene expression of *SKOR2*, *U2AF1* and *CASP3*, we conducted two complementary experiments. First, we infected THP-1 cells with three replication-competent HIV-2 isolates expressing the Vpx protein. This was then followed by RNA isolation and qPCR. Additionally, we transfected THP-1 cells with plasmids encoding HIV-2 Vpx and a functionally restricted version of the protein and carried out the qPCR analysis for the same genes. Gene expression was quantified using real-time qPCR, with *GAPDH* serving as the housekeeping gene.

In regard to experiments using replication-competent virions, a significant reduction in *U2AF1* expression was observed in cells infected with the HIV-2 1806 isolate (* *p* < 0.05), while *CASP3* expression showed a significant downregulation after infection with the HIV-2 1654 isolate (* *p* < 0.05); however, the expression of *SKOR2* remained unaffected across all HIV-2 primary isolates.

In transfected THP-1 cells, *U2AF1* expression was significantly reduced in cells transfected with both the wild-type and the functionally restricted protein (**** *p* < 0.0001), while no significant changes were observed in the cases of *SKOR2* and *CASP3* (Figure 4).

### 2.4. CASP3 Activity Measurements

To assess CASP3 activity, we utilized a colorimetric substrate and incubated it with lysates of cells transfected with wild-type and functionally restricted Vpx for 16 h. The activity was evaluated in comparison to the positive control, wherein apoptosis was induced with staurosporine. We found that there was a significant increase in CASP3 activity in cells transfected with wild-type Vpx, whereas the activity of CASP3 in cells transfected with a functionally restricted Vpx was comparable to that of the control (Figure 5).

## 3. Discussion

Monocytes and macrophages play a crucial role in innate immune responses. HIV has been shown to persist in these myeloid cells in blood and tissues of virologically suppressed individuals [32,33]. CD4 receptors and CCR5 co-receptors are abundantly expressed in monocytes, macrophages and even myeloid precursor cells in the bone marrow, which may be attributed to the monocytopenia experienced in people living with HIV (PLWHIV) [34]. During infection, HIV utilizes the host cell machinery to drive viral replication in order to establish infection. Viral accessory proteins, such as Vpx, are known to inhibit host cellular restriction factors [9]. Vpx is encoded only in the genome of HIV-2 and the lineages of the SIV/sm [9]. Besides its cardinal role in down-regulating SAMHD1 [5,12,13], Vpx was also shown to be pivotal to the efficient translocation of viral PIC [24].

In this study, we aimed to elucidate other potential functions of this small accessory protein, through exploring changes in the transcriptomic milieu of the cells induced by the expression of Vpx; additionally, we were curious as to whether or not the cytokine profile might also be affected by this protein.

One limitation of our study is the inability to explore the specific signaling pathways underlying the differential gene expression observed in Vpx-transfected cells. Our primary aim was to conduct a comprehensive pan-transcriptomic analysis to capture the overall cellular changes induced by Vpx, rather than dissecting individual pathways. While the Vpx-mediated degradation of SAMHD1 is a well-known mechanism and SAMHD1 itself can influence transcriptional regulation, our approach was not designed to distinguish the direct effects of Vpx from secondary effects resulting from intrinsic cellular pathways such as SAMHD1 downregulation. Future studies focusing on specific pathways and targeted mechanistic analyses will be necessary to fully elucidate the regulatory networks involved.

To detect the Vpx-induced transcriptomic changes, we utilized bioinformatic analysis to reveal DEGs induced by transfection with the HIV-2 Vpx-coding construct in THP-1 monocytes. Overall, 825 genes were found to be differentially expressed in the presence of wild-type Vpx.

It is of utmost significance to note that wild-type Vpx also induced the upregulation of protocadherin gamma subfamily C4 (PCDHGC4), caspase 3 (CASP3), U2 small nuclear RNA auxiliary factor 1 like 5 (U2AF1L5), NOP58 ribonucleoprotein (NOP58) and sorting nexin 14 (SNX14).

Protocadherins (PCDHs) have a wide spectrum of activities, including cell–cell adhesion [35] and the activation or inhibition of different signaling pathways, such as PYK2 and FAK tyrosine kinases [36,37] and mTOR pathways [38]. Moreover, they were also linked to the regulation of proliferation (Wnt/β-catenin signaling and PI3K/AKT-signaling) and apoptosis (NF-κB and DEPDC1-caspase signaling) [39]. We found that in the presence of Vpx, the expression of PXDHGC4 was augmented. Interestingly, PXDHGC4 was recently identified as a DEG in HIV-associated kidney cancer [40].

The expression level of CASP3 was also found to be elevated in Vpx-transfected THP-1 cells 16 h post-transfection. Multiple studies had shown that CASP3 is involved in the life cycle of HIV, most notably in the induction of apoptosis [41,42,43]. Virion-associated Vpr is required for Vpr-induced apoptosis in cells that are in G1 or M, but not in G2 or S phase. CASP3 influences the location of Vpr at the nuclear envelope, and thereby, augments Vpr-induced G2 arrest [41,42,43]. Vpu was found to increase the expression and activation of CASP3 through NF-κB inhibition [44,45]. While HIV-2 does not encode for Vpu, it was previously shown that Vpx is a broad inhibitor of NF-κB activation, including CASP3-mediated p65 cleavage [22], which might explain the increased expression of CASP3 in Vpx-transfected cells.

We wanted to further probe the alteration of CASP3 levels, using a colorimetric substrate in transfected THP-1 cells. Expectedly, CASP3 levels were significantly upregulated in cells transfected with the wild-type, functionally restricted Vpx, and even the mock vector. Interestingly however, the level of CASP3 upregulation was very high in case of cells transfected with the wild-type Vpx compared to the others.

CASP3, among others, was chosen to be validated by qPCR analysis at a later time-point using replication-competent virions, although, in this instance, analysis was carried out 48 h after infection. In those experiments, CASP3 was found to be significantly downregulated after infection with HIV-2 1654. While the median value was lower in the other HIV-2 isolates compared to the HIV-1 control, a statistical significance was not reached.

Discrepancies between qPCR and transcriptomic data may stem from technical and biological variations inherent to each method. While transcriptomic analysis methods like RNA-seq are highly sensitive to broad expression patterns and can detect subtle downregulation, qPCR’s sensitivity is more limited and may not capture these smaller fluctuations, especially for genes expressed at low levels. The downregulation of CASP3 in the case of the HIV-2 1654 isolate observed by qPCR was observed after two days, which could have been a result of a cellular compensatory mechanism activated after the earlier upregulation [46].

In the presence of Vpx, transcripts of CORO7-PAM16 readthrough (CORO7-PAM16), SKI family transcriptional corepressor 2 (SKOR2) and U2 small nuclear RNA auxiliary factor 1 (U2AF) were downregulated. Importantly, SKOR2 is a protein coding gene belonging to the Ski (Sloan Kettering Institute) and SnoN protein family, and functions as a negative regulator of TGF-β1 (transforming growth factor (TGF β)/SMAD (mothers against decapentaplegic) signaling pathway by inhibiting the SMAD-p300/CBP association and disrupting the formation of R-Smad/Smad4 complexes [47]. Several studies have associated TGF-β1/SMAD signaling pathway with HIV-1 infection [48,49,50]. The expression of TGFβ was found to be upregulated in HIV-1-infected monocytic cells [51], and elevated levels of TGFβ have been reported during acute infection with HIV-1 [50].

TGFβ plays a key role in the cell differentiation, proliferation and regulation of the immune system. SIV infection in rhesus macaques can be influenced by the SMAD-dependent pathway of TGF-β1, through the regulation of SMAD2 and SMAD3 activity and by the downregulation of SMAD7 [52]. In our study, we hypothesized that Vpx activates TGFβ signaling by downregulating the expression of the SKOR2 gene, which directly associates with SMAD2/3 and SMAD4. The degradation of Ski and SnoN proteins, such as SKOR2, induces the expression of TGF-β1, this is supported by the upregulation of other components of the TGFβ signaling pathway such as SMAD4 and TGFβ Receptor 1 (TGFBR1), which was evident in our analysis of DEGs. SKOR2 was also chosen for qPCR analysis; however, while the median value for the HIV-2 groups was lower than that of the control, this difference did not reach statistical significance. The considerable variability observed, as indicated by the large standard deviation, and outlier values may have masked the potential effect. Regrettably, we are not able to repeat this experiment to assess the effect more conclusively. Nonetheless, we carried out further experiments to confirm our hypothesis, and cytokine analysis showed an increased level of secreted TGF-β1 in the medium of cells transfected with Vpx-coding construct, compared to the mock control. The upregulation of TGF-β1 was indeed found to increase viral pathogenesis, promote HIV latency and induce apoptosis, in addition to suppressing innate immune responses [48,53]. TGF-β1 may also induce apoptosis by increasing the expression of apoptosis-inducing factor and CASP3 [54].

The increased production of cytokines such as IL-6, IL-1 and TNF-alpha in monocytes is associated with non-AIDS comorbidities in PLWHIV [55]. The aging of macrophage results in increased inflammation through the expression of proinflammatory cytokines (IL-6, IL-1) and the secretion of these cytokines results in tissue damage [56,57].

It was previously shown that HIV-1 Vpr can induce IL-6 production in monocyte-derived macrophages (MDMs) [58]. When we assayed for the level of cytokines in the medium of transfected cells, we found that Vpx resulted in the increase in proinflammatory cytokines (IL-6 and IL-1β) as well as TGFβ, which is in agreement with other researchers’ findings [59], where the level of secreted IL-6 and IL-1β were significantly increased in SIV and HIV-2 infected cells. This finding supports the role of HIV-2 Vpx in modulating the immune system.

While HIV-2 Vpr was found to activate the type I IFN signaling pathway in human DCs [60], we did not observe the potentiating effects of Vpx on the secretion of type I IFN in the medium of transfected cells.

During viral replication, HIV relies on alternative splicing to produce early regulatory proteins, Tat and Rev, which are produced from completely spliced transcripts [61]. U2AF splicing factors are known to utilize and interact at the 3’A3 splice site, where the HIV-1 Tat protein is produced [62]. HIV-1 3’splice site has suboptimal polypyrimidine tract leading to low splicing efficiency, which is enhanced and stabilized by the binding of U2AF factors recruited by splicing enhancers [62]. Mutations of the splicing enhancers have been shown to cause the severe loss of Tat mRNA expression and diminished virus replication in HIV-infected cells [63]. Moreover, multiple splicing was implicated in HIV-1’s latency, together with factors affecting transcription in CD4+ T cells from HIV infected patients [64]. In the absence of crucial cellular transcription factors, such as NF-κB, inadequate levels of Tat proteins have been associated with HIV latency [65].

In our analysis of DEGs, we observed that the expression level of U2AF1 was downregulated by Vpx. Real-time qPCR validated the downregulation of U2AF1 in the case of HIV-2 1806 compared to the HIV-1 control. Moreover, we carried out qPCR analysis on cells transfected with plasmids coding for the wild-type and a functionally restricted Vpx, and we found that both the wild-type and functionally restricted proteins downregulated U2AF1.

We hypothesized that the downregulation of this splicing factor might have implications on the level of expressed Tat protein. Western blot analysis in HEK-293T cells indeed hinted to the fact that in the presence of Vpx, the decreased expression of HIV-1 Tat was observed. When we repeated the experiment using a functionally restricted Vpx, we also found a significant downregulation of HIV-1 Tat expression.

On a transcriptional level, Vpx also resulted in the downregulation of MYD88, an innate immune signal transduction adaptor protein, which stimulates signaling molecules that activate NF-kB [66]. Thereby, this may result in the dampening of proviral transcription and possibly in the promotion of latency; however, more studies are needed to verify this.

This study, however, is not without limitations. Firstly, we utilized a pcDNA3.1-NeGFP vector for the artificial expression of Vpx in THP-1 cells; therefore, the over-expression of this protein may have occurred, which necessitates caution when comparing our findings to what occurs in the context of viral infection. Validation experiments utilizing replication-competent virions, however, show that our findings are indeed accurate, certainly in the case of U2AF1 and CASP3, and even though no significant change was observed in the case of SKOR2, all median values in cells infected with HIV-2 isolates were lower than that of the control. It is important, however, to note that qPCR measurements in infected cells were carried out after 48 h of incubation to ensure sufficient viral replication and integration, compared to qPCR analysis in transfection experiment, wherein the cells were collected after 16 h of transfection.

Our findings of DEGs were in comparison to a functionally restricted protein and a mock control, and hence, it is safe to say that the effects observed were as a result of Vpx itself, rather than over-expression thereof or of the backbone of the expression vector. Notably, our analysis revealed a lack of overlap between differentially expressed genes induced by wild-type Vpx and those influenced by wild-type/functionally restricted Vpx backbone genes. As anticipated, the backbone genes encoded proteins and signaling molecules governing diverse cellular functions, including, but not limited to cell migration, the maintenance of cell membrane integrity, transcriptional regulation, genetic and chromatin control, mitogenic activation, RNA assembly, packaging and intracellular transport (see Appendix A).

In conclusion, our transcriptomic analysis of cells transfected with HIV-2 Vpx revealed that many pathways are affected by this accessory protein; moreover, GO term analysis revealed that the complex network of cellular processes, including the negative regulation of viral process, splicososomal snRNP assembly, type I interferon signaling pathway, the regulation of transcription elongation from the RNA polymerase II promoter, the negative regulation of innate immune response, the positive regulation of interferon beta production and DNA-templated transcription elongation, was affected by Vpx. The effect on cellular cytokine milieu was also studied and showed an increase in key proinflammatory cytokines induced by this protein. The augmentation of TGF-β1/SMAD signaling pathway triggered by HIV-2 Vpx, possibly through the downregulation of SKOR2, is indeed a novel finding, which might lead to apoptosis, tissue damage, immunosuppression and the enhancement of viral infection. Moreover, the downregulation of HIV-Tat by Vpx that was observed in our experiments needs to be further characterized, as this might have an important implication in latency. We believe that this manuscript provides important information about the complex role played by HIV-2 Vpx in priming and taming the cellular environment to allow for the establishment of the infection.

## 4. Materials and Methods

### 4.1. Plasmids

We utilized pcDNA3.1-Vpx-NeGFP plasmid encoding for HIV-2 Vpx in conjugation with GFP (Genscript Biotech Corporation, Piscataway, NJ, USA). Vpx sequence was identical to that found in the HIV-2 ROD-based vector, HIV-2 CGP [67]. K68A and R70A mutations were implemented in the Vpx coding region to create the functionally restricted mutant Vpx coding pcDNA3.1-MutVpx-NeGFP vector [31]. pcDNA3.1-NeGFP devoid of the Vpx was used as mock control. pcDNA1-Tat coding for HIV-1 Tat was obtained from Addgene, a gift from Akitsu Hotta (Plasmid #138478) [68].

### 4.2. Transfection of THP-1 Cells for Transcriptomic Analysis

The THP-1 monocyte cell line (ATCC Number: TIB-202) was maintained in RPMI medium supplemented with 10% fetal bovine serum (FBS) (Thermo Fisher Scientific, Waltham, MA, USA), without antibiotics. On the day of transfection, cells were seeded into a 6-well plate (5 × 10^5^ cells/well) in 500 µL serum- and antibiotics-free OPTI-MEM (Thermo Fisher Scientific) and transfected with 5 µg of pcDNA3.1-Vpx-NeGFP, pcDNA3.1-MutVpx-NeGFP or pcDNA3.1-NeGFP plasmids using Lipofectamine LTX reagent (Thermo Fisher Scientific), according to the manufacturer’s protocol. To verify expression of the constructs, an Evos FLoid cell imaging station (Thermo Fisher Scientific) and flow cytometry were used to analyze positively fluorescing cells.

### 4.3. Isolation of RNA for Transcriptomic Analysis

Cells were incubated at 37 °C, 5% CO_2_ for 16 h after transfection with 5 µg of pcDNA3.1-Vpx-NeGFP, pcDNA3.1-MutVpx-NeGFP or pcDNA3.1-NeGFP plasmids, and were then washed in 1 mL ice-cold PBS twice. Total RNA was extracted using TRIzol Reagent (Thermo Fisher Scientific), according to the manufacturer’s instructions. Experiments were carried out in duplicates.

### 4.4. RNA-Seq Library Preparation and Sequencing

Total RNA sample quality was verified by Agilent 2100 BioAnalyzer using the Agilent RNA 6000 Nano Kit (Agilent Technologies, Waldbronn, Germany), according to manufacturer’s protocol. Samples with RNA integrity number (RIN) value > 7 were accepted for library preparation process. RNA-Seq libraries were prepared from total RNA using an Ultra II RNA Sample Prep kit (New England Biolabs, Ipswich, MA, USA) according to the manufacturer’s protocol. Poly-A RNAs were captured by oligo-dT conjugated magnetic beads then the mRNAs were eluted and fragmented at 94 °C. First strand cDNA was generated by random priming reverse transcription and after second strand synthesis step, double stranded cDNA was generated. After repairing ends, A-tailing and adapter ligation steps, adapter ligated fragments were amplified in enrichment PCR, and finally, sequencing libraries were generated. Single-end 75 cycles sequencing was completed on Illumina NextSeq 500 platform (Ilumina, San Diego, CA, USA). Library preparation and sequencing were performed at Genomic Medicine and Bioinformatics Core Facility of the University of Debrecen, Hungary.

### 4.5. RNA-Seq Data Processing and Analysis

We assessed the quality of individual sequences using the FastQC software (v0.11.9) [Babraham Institute, https://www.bioinformatics.babraham.ac.uk/projects/fastqc/] (URL accessed on 20 February 2023) prior to and following trimming with Trimmomatic (v0.39) [69]. The mean of per base quality scores in raw sequences fell into the best quality region with a widening range of values in the last third of the read fragments, while surpassing 35 across the last 70 bases after filtering (Appendix A). The Q30 Phred quality scores (i.e., 99.9% probability of a base call being correct) were >95% as calculated by fastp (v0.23.2) [70] from the raw fastq files of all samples. After removing lower quality and adapter sequences, we calculated the means and standard deviations of the resulting fragment lengths for each sample and defined them to control the distribution by the fragment/length options for the accuracy of expression level estimates from single-end data in RSEM (v1.3.3). Overall, trimming resulted in a mean fragment length of 66 with a standard deviation of 21 bases. From the RSEM workflow, we ran Bowtie 2 (v2.4.4) [71] with the default 25 seed length parameter to align reads to the GRCh38.p13 human reference genome. The expression metric provided by RSEM is an estimate of the number of fragments derived from a gene, as reads often map to more than a unique region. This count is frequently a non-integer value, which represents the number of alignable fragments from a gene based on its maximum likelihood (ML) abundances [72]. We computed the alignment quality metrics using MultiQC (v1.11) [73]. Sequencing depth reached the suggested 10 M total number of aligned reads for RNA-Seq experiments in all samples [74]. The count matrices of the gene-level estimates generated by RSEM were further processed in the R environment (v4.2.2) [R Core Team (2022). R: A Language and Environment for Statistical Computing. R Foundation for Statistical Computing, Vienna, Austria. https://www.R-project.org/]. For differential gene expression analysis, we applied the EBSeq package (v1.36.0) [75]. Alterations in gene expression levels of the three sample types (GFP control, wild-type and mutant Vpx) were evaluated in one comprehensive multiple condition comparison model to exclude latent expression patterns not related to the studied conditions. With these three conditions, there are five possible expression patterns, as shown in Table 1.

### 4.6. Gene Ontology Enrichment Analysis and Functional Annotation Analysis

Gene annotation was retrieved from the daily updated ‘gene_info’, ‘gene2go’, ‘hiv_interactions’ and ‘gene2pubmed’ files available on the NCBI FTP site (https://ftp.ncbi.nlm.nih.gov/gene/DATA/) (URL accessed on 16 January 2024) and reorganized by in-house shell scripts. We carried out enrichment analysis for Gene Ontology (GO) with the topGO R package (v2.48.0) using a custom-generated gene-to-GOs mapping file. The gene universe consisted of the entire set of genes detected in all samples, and significant genes were filtered according to a *p*-value < 0.05 threshold (to obtain *p*-values corresponding to classical statistics, we applied the 1-PPDE transformation). In the topGO algorithm we chose the default ‘weight01’ method with the Kolmogorov–Smirnov statistical test and selected the enriched GO terms with FDR-adjusted *p*-values ≤ 0.05 with at least 10% DEGs of all annotated genes where the total size was less than 1000 genes. The interpretation of the results was facilitated by collecting the HIV1 interaction partners coded by the DEGs, and PubMed metadata mining from publications linked to the DEGs with keywords related to any type of viral infection. Data visualization was performed with ggplot2 (v3.3.6), ggrepel (v0.9.1), GOplot (v1.0.2) and formattable (v0.2.1) R packages.

### 4.7. Activation and Transfection of THP-1 Cells for Cytokine Measurements

A total of 500,000 cells per well in triplicates were plated onto 6-well plates in 500 µL RPMI-1640 medium (Thermo Fisher Scientific) containing 10% fetal bovine serum (FBS) and 1% L-glutamine (Thermo Fisher Scientific). After 3 h of incubation at 37 °C in 5% CO_2_, THP-1 cells were activated with 100 nM phorbol 12-myristate 13-acetate (PMA) (Thermo Fisher Scientific) followed by incubation for 1 h. The PMA-containing medium was replaced with 500 µL fresh RPMI medium and the cells incubated for 24 h. Optical microscope was used to verify the differentiation of monocytes from adherent macrophages. The activated cells were transfected with 5 µg of pcDNA3.1-Vpx-NeGFP or pcDNA3.1-NeGFP (mock) vectors using Lipofectamine LTX Reagent in Opti-MEM (Thermo Fisher Scientific) and incubated for 5 h at 37 °C, 5% CO_2_. Cell control and native control (containing lipofectamine transfection reagent only) were included. Thereafter, the supernatant was replaced with 1 mL of fresh RPMI-1640 medium followed by incubation for 24 h. The centrifugation of the cells was performed for 3 min at 11,000× *g*, and the medium was collected for cytokine measurements. The cytokine levels of supernatants (IL-1β, TGF-β, IL-6 and IFN-α) were determined using ELISA kits (BD Biosciences, San Diego, CA, USA) according to the manufacturer’s instructions. The quantification of interferon alphas (IFNAs) was determined using the Human IFN-Alpha ELISA Kit (TCM) (PBL Assay Science, Piscataway, NJ, USA), also following the manufacturer’s protocol. The comparison of cytokine levels in different transfection conditions were performed with non-parametric Kruskal–Wallis test using GraphPad Prism 7.0 (GraphPad software, La Jolla, CA, USA).

### 4.8. Detection of Vpx Effect on the Expression of HIV-1 Tat by Western Blot

Western blot analysis was used to confirm the expression of HIV-1 Tat protein in the presence of Vpx. The day before transfection, 1 × 10^6^ HEK-293T cells (Invitrogen, Carlsbad, CA, USA) were seeded into T-25 flasks in 5 mL Dulbecco’s Modified Eagle’s Medium (DMEM) (Sigma-Aldrich, St. Louis, MO, USA) supplemented with 10% FBS, 1% glutamine and 1% penicillin-streptomycin. On the day of transfection, cells were transfected with 5 µg of pcDNA3.1-Vpx-NeGFP or pcDNA3.1-NeGFP (mock) vectors using the polyethylenimine (PEI) (Sigma-Aldrich, St. Louis, MO, USA). PEI-treated, non-transfected cells were used as controls. On the following day, cells were transfected again with 5 µg of pcDNA1-Tat plasmid encoding HIV-1 Tat, using PEI. After 24 h of incubation, the supernatant was discarded and cells were washed with 1 mL of PBS and gently scraped off the flask. Cells were centrifuged for 5 min at 870 RPM RT, and the pellet was washed with 5 mL, then with 1 mL, of PBS.

The pellet was then re-suspended in 300 µL of Lysis buffer (50 mM Tris-HCl, 250 mM of NaCl, 5 mM of EDTA, 50 mM of NaF, 0.5% of NP-40, pH 7.4) and was thereafter incubated for 30 min on ice and vortexed for 5 s every 10 min. After incubation, samples were sonicated (Realsonic sonicator, 10 s at 40% energy at 4 °C) and centrifuged for 30 min at 14,000× *g* at 4 °C. Supernatant was then transferred into a clean tube and a Pierce BCA protein assay kit (Thermo Scientific, Waltham, MA, USA) was used to determine the protein concentration. A total of 15 µg of protein from samples was loaded onto 18% SDS-acrylamide gel, then proteins were blotted onto a nitrocellulose membrane (Biorad, Hercules, CA, USA). Tat antiserum (NIH AIDS Reagent Program, Division of AIDS, NIAID, NIH, Bethesda, MD, USA) and anti-β-actin (Sigma-Aldrich, St. Louis, MO, USA) were used as primary antibodies, followed by anti-rabbit (Biorad, Hercules, CA, USA) and anti-mouse (Sigma-Aldrich, St. Louis, MO, USA) as secondary antibodies. Blots were detected using WesternBright PICO substrate (Advansta, San Jose, CA, USA) and Azure600 GelDoc (Dublin, CA, USA).

### 4.9. Infection of THP-1 Cells

Three subtype A HIV-2 primary isolates were used. All isolates (1010, 1654 and 1806) originated from individuals from West Africa [76]. HIV-1_IIIB_ was included as a control in all experiments. Virus stocks were prepared by infecting 5 × 10^6^ phytohemagglutinin P (Pharmacia, Uppsala, Sweden)-stimulated human peripheral blood mononuclear cells (PBMCs) from healthy blood donors. Fresh PBMC was added once a week to the virus cultures and supernatants were harvested on days 7, 14 and 21 after infection and stored at −80 °C until use. Before viral infection, 250,000 THP-1 cells per well in quadruplicates, were plated into 24-well plates in 500 µL of RPMI-1640 medium (Thermo Fisher Scientific) containing 10% fetal bovine serum (FBS) and 1% L-glutamine (Thermo Fisher Scientific). After 3 h of incubation at 37 °C in 5% CO_2_, THP-1 cells were activated with 100 nM phorbol 12-myristate 13-acetate (PMA) (Thermo Fisher Scientific) followed by incubation for 1 h. The PMA containing medium was then replaced with 500 µL of fresh RPMI medium and the cells were incubated for further 24 h. An optical microscope was used to verify the differentiation of monocytes to adherent macrophages. The activated cells were then infected with 25 ng of capsid antigen equivalents of HIV-1 or HIV-2 isolates and incubated for 48 h at 37 °C, 5% CO_2_. Native cell control (non-stimulated with PMA) and PMA-stimulated cell controls were included. Infection was confirmed by the measurement of HIV-1 or HIV-2 RNA in cell lysates using RT-qPCR specific for HIV-1 and HIV-2, respectively, as previously described [77].

### 4.10. Real-Time qPCR for SKOR2, U2AF1 and CASP3 Genes

RNA was isolated from infected THP-1 cells using the RNeasy Mini Kit (Qiagen, Hilden, Germany), following the manufacturer’s protocol. cDNA synthesis was performed on extracted RNA using SuperScript IV Reverse Transcriptase (Thermo Fisher Scientific, Waltham, MA, USA), R1 primer (5′-GGT CAT CAT CAT Cwr mAT CTA yAT C-3′) and RiboLock RNase inhibitor (Invitrogen, Taastrup, Denmark) at an initial 5 min incubation at 23 °C, followed by 10 min at 55 °C, and a final heat-inactivated step at 80 °C for 10 min. Quantitative real-time PCR was performed using the SYBR Select Master Mix (Thermo Fisher Scientific, Waltham, MA, USA) and gene-specific primers (Appendix A), utilizing CFX96 Touch Real-Time PCR machine (Bio-Rad, Hercules, CA, USA), and the results were analyzed using the ΔΔCt method. Statistical analysis was performed with non-parametric Kruskal–Wallis test using GraphPad Prism 7.0 (GraphPad software, La Jolla, CA, USA).

### 4.11. Transfection of THP-1 and RNA Isolation for qPCR

To analyze the effects of Vpx on the expression of U2AF1, SKOR2 and CASP3 genes, 500,000 THP-1 cells were plated in a six-well plate in full RPMI medium containing 10% FBS and 1% L-glutamine in triplicates. The cells were transfected with 5 µg of pcDNA3.1-Vpx-NeGFP, pcDNA3.1-MutVpx-NeGFP or pcDNA3.1-NeGFP plasmids in lipofectamine followed by 16 h incubation at 37 °C and 5% CO2. The cells were collected in 500 µL TRIzol Reagent (Thermo Fisher Scientific) and RNA isolation was carried out according to the manufacturer’s instructions. Quantitative real-time qPCR was carried out using a 500 ng RNA template, Cells-to-CT TaqMan RT-qPCR-master mix (Thermo Fisher Scientific), TaqMan gene specific expression assay for SKOR2 (Gene ID- Hs04400272_m1), U2AF1 (Gene ID—Hs00739599_m1) and CASP3 (Gene ID—Hs00234387) and GAPDH (Hs02786624_g1), using Light Cycler LC480_384 machine (Roche, Hoffmann Ltd., Switzerland). RT-PCR conditions were as follows: reverse transcription 50 °C for 5 min (1 cycle), RT inactivation for 95 °C for 15 s (1 cycle) and amplification at 95 °C for 15 s and 60 °C for 1 min (40 cycles), then calculations were conducted using the ΔΔCt method. Statistical analysis was performed using non-parametric *t*-test and analyzed by GraphPad Prism 7.0 (GraphPad software, La Jolla, CA, USA).

### 4.12. Caspase 3 (CASP3) Activity Measurement

In a 6-well plate, 500,000 THP-1 cells/well in triplicates were seeded in antibiotic-free RPMI medium enriched with 10% fetal bovine serum (FBS) and 1% L-glutamine (Thermo Fisher Scientific) to a final volume of 500 µL per well. Using Lipofectamine LTX reagent (Thermo Fisher Scientific), the cells were transfected with 5 µg of pcDNA3.1-Vpx-NeGFP, pcDNA3.1-MutVpx-NeGFP or pcDNA3.1-NeGFP (mock). For the positive control, the cells were treated with 1 µM of staurosporine (MedChem Express (MCE), Monmouth Junction, NJ, USA) to induce apoptosis, additionally, a native control containing lipofectamine LTX-only treated cells was included. Following incubation at 37 °C, 5% CO_2_ for 16 h, the cells were collected and lysed in a buffer containing 10 mM of Tris, 1 mM of dithiothreitol, 2 mM of EDTA and 1 mM of PMSF [pH 7.4] for 30 min on ice. The lysates were centrifuged at 14,000 rpm for 30 min at 4 °C and protein concentration was determined by BCA. CASP3 activity was measured using MedChem Ac-DEVD-pNA colorimetric substrate (CAS no. 189950-66-1, MedChem Express), following the protocol that was described previously [78]. In a 96-well plate, 0.2 mM of Ac-DEVD-pNA substrate diluted in the Lysis buffer, was added to 40 µg of cell lysate to achieve a final reaction of 100 µL per well. The preparation was incubated for 1 h at 37 °C and absorbance was thereafter measured at 405 nm using an Agilent BioTek Synergy H1 microplate reader (Agilent Technologies, Santa Clara, CA, USA).

## Figures and Tables

**Figure 1 ijms-26-03460-f001:**
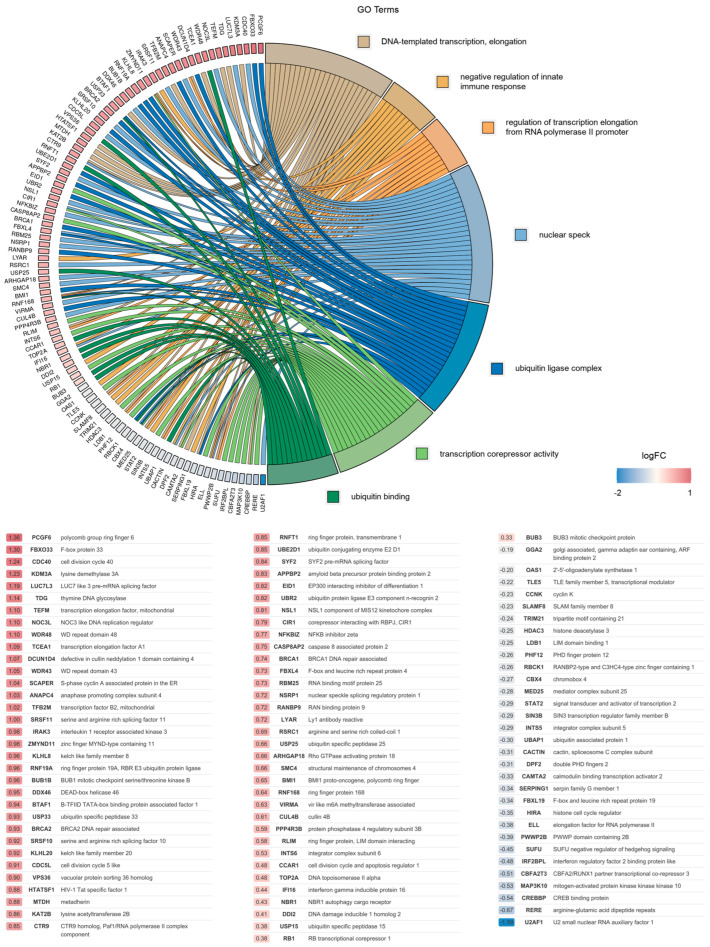
Gene association analysis of DEGs.

**Figure 2 ijms-26-03460-f002:**
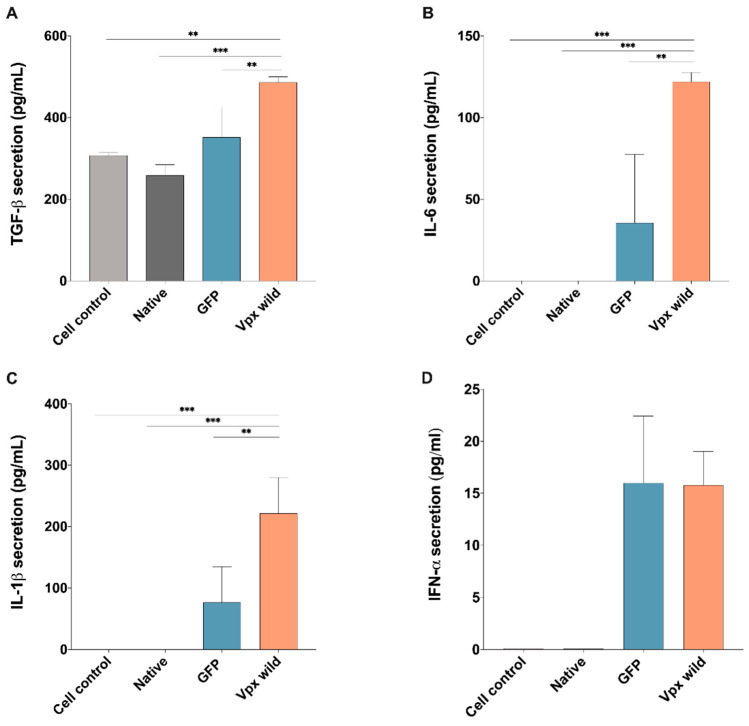
HIV-2 Vpx effect on the secretion of proinflammatory cytokines and type I interferons. Secretion of (**A**) transforming growth factor TGF-β (TGFβ), (**B**) Interleukin-6 (IL-6) and (**C**) Interleukin-1 beta (IL-1β) and (**D**) type I α interferons determined from the supernatant of Vpx-transfected cells, 24 h after transfection. Columns represent the mean values of two independent experiments, and error bars represent the ± SD of these experiments. ** *p*  <  0.01; *** *p*  <  0.001.

**Figure 3 ijms-26-03460-f003:**
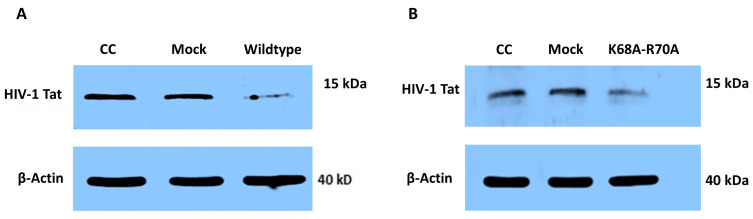
Detection of HIV-1 Tat expression in the presence of Vpx by Western blot. (**A**) HEK-293T cells were transfected with pcDNA3.1-Vpx-NeGFP or pcDNA3.1-NeGFP constructs followed by transfection with pcDNA1-Tat plasmid encoding for HIV-1 Tat. Western blot of cell lysate was carried out 24 h after transfection with pcDNA1-Tat to determine the effect of Vpx on the expression of Tat. CC: cells transfected with only pcDNA1-Tat; Mock: cells transfected with pcDNA3.1-NeGFP followed by transfection with pcDNA1-Tat. (**B**) The same experiment was carried out using a functionally restricted Vpx (K68A-R70A mutant). kDa: Kilodalton. β-actin was used for normalization.

**Figure 4 ijms-26-03460-f004:**
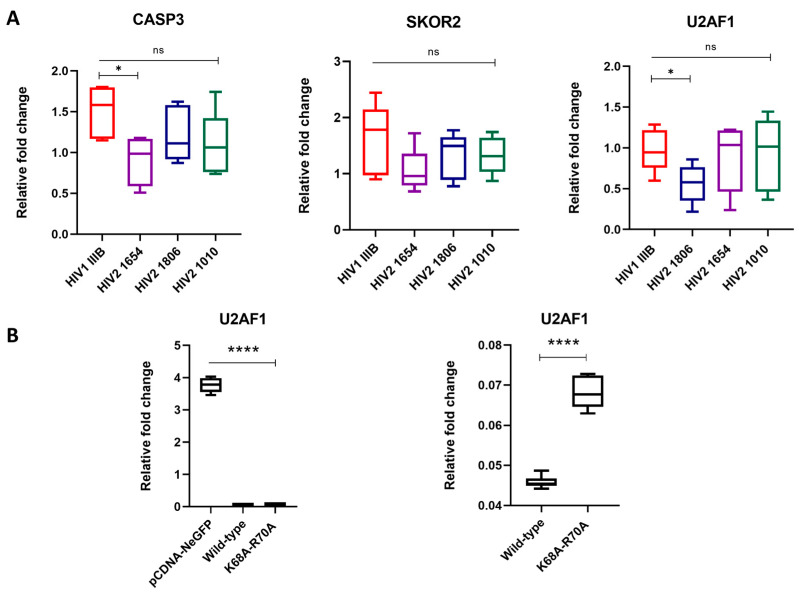
RT-qPCR validation of significantly altered transcripts. Relative fold changes in the gene expression of SKOR2, CASP3 and U2AF1 in qPCR results from (**A**) THP-1 cells infected with replication-competent HIV-1 and HIV-2 isolates (25 ng capsid antigen equivalents) for 48 h, whereafter infection was confirmed by the detection of viral RNA in cell lysates, and (**B**) THP-1 cells transfected with wild-type and functionally restricted Vpx. pcDNA-NeGFP was used as a mock control. * *p* < 0.05; **** *p* < 0.0001.

**Figure 5 ijms-26-03460-f005:**
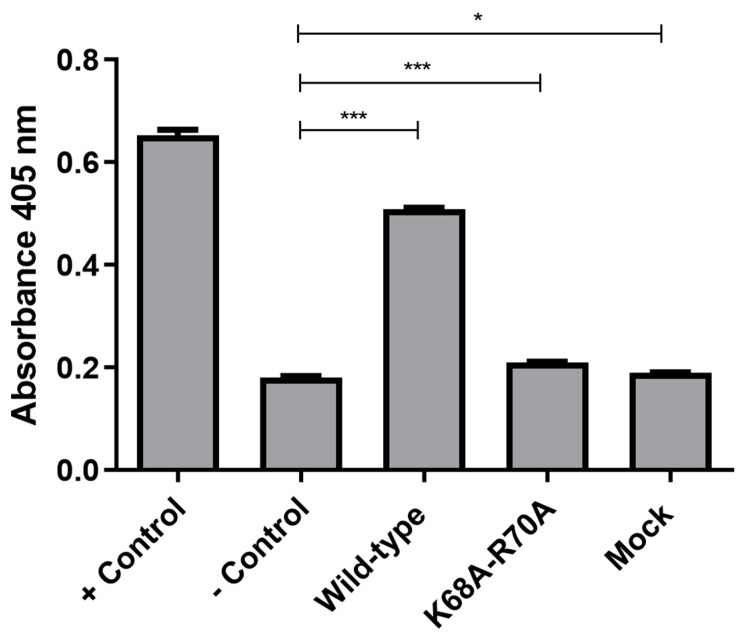
Assessment of CASP3 activity in transfected cells. CASP3 activity is shown in cells transfected with pcDNA3.1-Vpx-NeGFP (wild-type), pcDNA3.1-MutVpx-NeGFP (K68A-R70A mutant) and pcDNA3.1-NeGFP (mock) plasmids. Non-transfected THP-1 cells were used as negative controls (− Control), while cells treated with 1 µM staurosporine were used as positive controls (+ Control). An unpaired *t*-test was used to calculate statistical significance. * *p*-value ≤ 0.05; *** *p*-value ≤ 0.001. The experiment was carried out in triplicates.

**Table 1 ijms-26-03460-t001:** Gene expression patterns in sample conditions.

*Pattern*	*Relation of Expression Levels*
*1*	$q_{g_{i}}^{GFP}=q_{g_{i}}^{MUT}=q_{g_{i}}^{WT}$
*2*	$q_{g_{i}}^{GFP}=q_{g_{i}}^{MUT}\neq q_{g_{i}}^{WT}$
*3*	$q_{g_{i}}^{GFP}=q_{g_{i}}^{WT}\neq q_{g_{i}}^{MUT}$
*4*	$q_{g_{i}}^{GFP}\neq q_{g_{i}}^{MUT}=q_{g_{i}}^{WT}$
*5*	$q_{g_{i}}^{GFP}\neq q_{g_{i}}^{MUT}\neq q_{g_{i}}^{WT}$

Where *q*^GFP^(*g_i_*) is interpreted as the expression quantity (*q*) of gene i (*g_i_*) in a given sample type, e.g., GFP control. EBSeq—with a Bayesian statistical approach—computes the posterior probabilities of genes being differentially expressed (PPDE) in a particular pattern. Genes with PPDE ≥ 0.95 are considered differentially expressed (DEGs) in a significant manner with target false discovery rate (FDR) controlled at α = 0.05. To obtain DEGs in wild type, we selected significant genes from patterns 2 and 5 and the mutants from patterns 3 and 5. Pattern 4 contains genes whose expression changed due to the burden of viral protein production imposed on the protein translation machinery in both wild-type and mutant samples.

## Data Availability

The RNA-Seq data are available under NCBI GEO accession no. GSE233999. For reviewing purpose, the following token has been generated to grant reviewers access to the database (cpgdeowgflcjfop), accessible through https://www.ncbi.nlm.nih.gov/geo/query/acc.cgi?acc=GSE233999 (accessed date: 24 January 2024), 2025.

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
