# Peer review of "Transcriptomic Analysis Reveals Key Pathways Influenced by HIV-2 Vpx"

_ijms, 2025, doi:10.3390/ijms26083460_

Round 1
Reviewer 1 Report
Comments and Suggestions for Authors
Ilona Szojka et al have conducted a transcriptomic analysis of THP-1 cells transfected with constructs expressing Wt and mutant versions of the HIV-2 accessory protein Vpx. They demonstrate differential expression of pathways involved in regulation ofn the innate immune response, as well as induction of pro-inflammatory cytokines. They further show co-transfection of Vpx and HIV-1 Tat results in down-regulation of the latter.
Major comments
Overall, the study is well written and presented. Some concerns however there are some areas that need to be addressed. Firstly, given the major roles of Vpx are the degradation of SAMHD1 and the HUSH complex, are the transcriptomic changes observed simply due to the loss of these two factors. Indeed, Bonifati et al note that loss of SAMHD1 in THP-1 cells results in substantial changes in apoptosis, cell cycle status and cell proliferation (PMID: 27183329). Furthermore recent work has shown that SAMHD1 plays a key role in DNA repair pathways (PMID: 28834754, PMID: 36344525) as well as controlling the homeostatic balance of the cellular dNTP pool. There is no discussion of these rather pertinent roles of Vpx which would result in significant impacts on the cellular transcriptome. In light of the relatively minimal impact of the catalytically inactive Vpx mutant had on gene expression of THP-1 further emphasises that regulation of either SAMHD1 or the HUSH complex contributes to the changes in gene expression. The justification for the transcriptomic analysis is no clear, if the goal was to identify additional roles of Vpx beyond what is canonically understood, then these experiments should be repeated in SAMHD1 or HUSH KO cells.
Additionally, the level of reported Vpx expression in these is concerning where only 50-60% of sequenced cells expressed the protein. This data should be included in the manuscript. Sequencing of these bulk samples will result in a mixed population o expressing and non-expressing cells reducing the robustness of the analysis and conclusions. Generation of stable cell lines or sorting of Vpx expressing cells before sequencing would have been a more appropriate approach.
Concerning the interaction between Tat and Vpx, additional mechanistic experiments should be performed to validate these findings. Is this also seen with the catalytically inactive mutant for example? Does inhibition of the proteosome rescue this phenotype? Is this also seen in a full infection model? Does Vpx directly interact with Tat?
Regarding the CASP3 experiments, reports by Davenne et al show that loss of SAMHD1 result in cells being more susceptible to apoptosis due to disruption of cellular dNTP pools (PMID: 32402273). Vpx-mediated degradation of SAMHD1 is a likely explanation for the observed phenotype. Further these experiments would benefit from validation of Vpx expression by western blot of FACS.
Finally viral parameters confirming infection are missing from figure 6 limiting the interpretation of these data.
Author Response
Major comments
- Overall, the study is well written and presented. Some concerns however there are some areas that need to be addressed. Firstly, given the major roles of Vpx are the degradation of SAMHD1 and the HUSH complex, are the transcriptomic changes observed simply due to the loss of these two factors. Indeed, Bonifati et al note that loss of SAMHD1 in THP-1 cells results in substantial changes in apoptosis, cell cycle status and cell proliferation (PMID: 27183329). Furthermore recent work has shown that SAMHD1 plays a key role in DNA repair pathways (PMID: 28834754, PMID: 36344525) as well as controlling the homeostatic balance of the cellular dNTP pool. There is no discussion of these rather pertinent roles of Vpx which would result in significant impacts on the cellular transcriptome. In light of the relatively minimal impact of the catalytically inactive Vpx mutant had on gene expression of THP-1 further emphasises that regulation of either SAMHD1 or the HUSH complex contributes to the changes in gene expression. The justification for the transcriptomic analysis is no clear, if the goal was to identify additional roles of Vpx beyond what is canonically understood, then these experiments should be repeated in SAMHD1 or HUSH KO cells.
- Response:
Dear Reviewer, thank you very much for the time and effort spent reviewing our manuscript. We will hereby try our best to address your comments and concerns. We understand the main concern raised, and would like to clarify that our study was primarily focused on investigating the direct effects of Vpx transfection on the transcriptomic profile of THP-1 cells, particularly comparing the wild-type and functionally restricted K68 and R70 mutant Vpx. While it is acknowledged that Vpx is known to target SAMHD1 for degradation, our experimental design intentionally centered on assessing the transcriptomic changes induced by the Vpx proteins themselves. Although SAMHD1 downregulation may be one of the downstream consequences of Vpx transfection, our primary aim was not to investigate SAMHD1’s direct involvement in the transcriptomic changes but rather to explore the global gene expression alterations triggered by Vpx. Therefore, we did not specifically focus on SAMHD1 levels or other potential downstream factors such as the HUSH complex. That said, we recognize that SAMHD1 downregulation could indeed influence the transcriptome and warrant consideration. From other studies, we know that the mechanism of SAMHD1 expression or degradation is influenced by other factors or proteins as reviewed by (PMID: 32244340). When searching our data for downstream players in the SAMHD1 pathway, indeed some transcripts of genes related to the innate immune response and interferon simulated genes were upregulated (IFIH1), perhaps the downregulation of SAMHD1 by Vpx resulted in the triggering of cGAS-STING and RIG-I/MDA5 pathways, leading to increased IFN response. Other transcripts of genes that are also connected to SAMHD1 pathway were also upregulated (RRM2 and DCK) in case of cells transfected with the wildtype VPX, hinting to linkage to the downregulation of SAMHD-1 by Vpx.
The introduction has now the following additions: “SAMHD1 is abundant in cells of the myeloid lineage [10], however, previous studies have indicated reduced basal levels of SAMHD1 in monocyte-derived macrophages [5,11]”
and: “It is worth mentioning that other viruses, besides HIV, were found to counteract SAMHD1, either directly through post-translational modifications, such as phosphorylation by herpes viruses [16], or indirectly modulating its activity, such as in the case of HTLV-1 and HBV [17,18]”
The following paragraph is now added to the discussion section: “One limitation of our study is the inability to explore the specific signaling pathways underlying the differential gene expression observed in Vpx-transfected cells. Our primary aim was to conduct a comprehensive pan-transcriptomic analysis to capture the overall cellular changes induced by Vpx, rather than dissecting individual pathways. While Vpx-mediated degradation of SAMHD1 is a well-known mechanism, and SAMHD1 itself can influence transcriptional regulation, our approach was not designed to distinguish between direct effects of Vpx and secondary effects resulting from intrinsic cellular pathways such as SAMHD1 downregulation. Future studies focusing on specific pathways and targeted mechanistic analyses will be necessary to fully elucidate the regulatory networks involved“
We hope this clarification helps, and we appreciate the opportunity to refine our manuscript.
- Comment 2: Additionally, the level of reported Vpx expression in these is concerning where only 50-60% of sequenced cells expressed the protein. This data should be included in the manuscript. Sequencing of these bulk samples will result in a mixed population o expressing and non-expressing cells reducing the robustness of the analysis and conclusions. Generation of stable cell lines or sorting of Vpx expressing cells before sequencing would have been a more appropriate approach.
- Response:
Thank you for this observation, this was a mistake on our side, we re-analyzed the data and saw that the transfection efficiency was higher than the stated one. Attached is a figure of GFP fluorescence for the reviewer (Transfection efficiency for Reiewer 1). Our experiments utilized a functionally restricted Vpx in addition to a mock control, therefore we are very confident that the results obtained are a result of transfection with Vpx.
- Comment 3: Concerning the interaction between Tat and Vpx, additional mechanistic experiments should be performed to validate these findings. Is this also seen with the catalytically inactive mutant for example? Does inhibition of the proteosome rescue this phenotype? Is this also seen in a full infection model? Does Vpx directly interact with Tat?
- Response:
Thank you for this insight. We performed a western blot of lysate of cells that were transfected with a functionally restricted (K68 and R70 mutant) Vpx, and found that downregulation of HIV-1 Tat was also noticed in this case, compared to cells transfected with the mock control. We agree with the reviewer that further exploration of the mechanistic events leading to the downregulation of Tat by Vpx needs to be explored further, however, at the moment; we feel that this would be better served by follow up studies. As stated, in the manuscript, we hypothesized that downregulation of U2AF splicing factor might have had implications on the level of expressed Tat protein, given that in our previous study (PMID: 29743354) we noticed a downregulation of HIV-1 replication by HIV-2, and thought that it might have something to do with Vpx. That is the reason we probed Tat in this case, but as stated by the reviewer, a thorough mechanistic insight would be of great benefit in a follow-up study.
- Comment 4: Regarding the CASP3 experiments, reports by Davenne et al show that loss of SAMHD1 result in cells being more susceptible to apoptosis due to disruption of cellular dNTP pools (PMID: 32402273). Vpx-mediated degradation of SAMHD1 is a likely explanation for the observed phenotype. Further these experiments would benefit from validation of Vpx expression by western blot of FACS.
- Response:
We agree with the reviewer that downregulation of SAMHD1 might have an effect on apoptosis as has been previously documented, however, since analysis of downstream effects of SAMHD1 was not our primary aim, we did not attempt to elaborate on it. A mention of this has now been added to the introduction and discussion sections.
- Comment 5: Finally viral parameters confirming infection are missing from figure 6 limiting the interpretation of these data.
- Response:
For clarification the legend of Figure 6 has been revised, now pointing to inoculum virus dose (25 ng capsid antigen equivalents) and confirmation of infection by detection of viral RNA in cell lysates. In line with above clarifications the 4.9. section in the Material and Methods has also been revised.
Reviewer 2 Report
Comments and Suggestions for Authors
The manuscript titled “Transcriptomic analysis reveals key pathways influenced by HIV-2 Vpx” by Zsófia Ilona Szojka et al., investigates the analysis of deferentially expressed genes upon overexpression of HIV-2 accessory protein viral protein X. The authors carried out transcriptomic analysis of THP-1 cells ectopically expressing wild type and functionally restricted mutant of HIV-2 Vpx. The authors validated the expression of significantly altered genes by qPCR confirming the role of Vpx in regulating Key pathways. The manuscript is presented in clear and concise language with significant findings. I have only one query regarding their transcriptomic analysis. Authors proposed Vpx facilitates the expression of HIV-1 Tat by downregulating the crucial splicing factors. The authors did WB analysis of ectopically expressed HIV-1 Tat upon Vpx overexpression in HEK-293T cells (Figure-5). Did the authors try knocking down of any of these top hits (such as U2AF1) and check if that has similar effect on HIV-1 Tat expression?
Author Response
- Comment 1: Authors proposed Vpx facilitates the expression of HIV-1 Tat by downregulating the crucial splicing factors. The authors did WB analysis of ectopically expressed HIV-1 Tat upon Vpx overexpression in HEK-293T cells (Figure-5). Did the authors try knocking down of any of these top hits (such as U2AF1) and check if that has similar effect on HIV-1 Tat expression?
- Reponse:
Dear reviewer, thank you for the time and effort spent reviewing our manuscript. In a previous study (PMID: 29743354) we noticed a downregulation of HIV-1 by HIV-2, and Vpx was involved in this downregulation. In this study, when we observed that there was a downregulation of a crucial splicing factor (U2AF) for Tat as a result of transfection with Vpx, we immediately hypothesized that this might exert an effect on Tat expression. We have repeated the western blot experiment from lysate of cells transfected with the functionally restricted Vpx (K68 and R70 mutant), and noticed that downregulation of Tat expression was also observed in this case, although not quite as marked as in the case of WT Vpx. The manuscript has now been supplemented with this information. We agree with the reviewer that the mechanism of this downregulation needs to be explored further, however, we believe that this would be the subject of a future complementary study, as the main aim of this manuscript is a pan-transcriptomic analysis of the effects of Vpx in THP-1 cells.
Reviewer 3 Report
Comments and Suggestions for Authors
This manuscript aims to elucidate the changes in transcriptomic milieu and cytokine profile induced by Vpx through cell lines. Vpx is a virus-associated accessory protein that is essential for infection and play a crucial role in regulating the immune response of host cells.
Main weakness: The presentation of Results Sections 2.1 and 2.2 is long-winded and difficult for the reader to follow, undermining the main subject of this manuscript.
Suggestion: Rewrite it and combine them together. Move the figure 1, figure 2 and table 1 to supplemental data. Keep the figure 3, figure 4 and make everything simple. Point the most important finding is enough.
Minors:
- Abbreviation needs to be fully presented at the first appearance in the manuscript. Make corrections at the final version.
- Line 129: “mucin 4, cell surface associated (MUC4)”. Is it a wrong presentation in text?
- Line 150: “mal, T cell differentiation protein 2 (MAL2)”. Is it a wrong presentation in text?
- Line 276-284: bolted.
- Line 299: “Viral accessory proteins; such as Vpx, are” Is “;” a typo err of “,”?
- Line 317: “asPYK2” needs a space.
- Line 312-313: “protocadherin gamma subfamily C, 4 (PCDHGC4)” delete the “,”
- Line 590: Change the “,” into “.”
By the way, the iThenticate similarity of this manuscript is 75% that is super high. Is this manuscript published in another place?
Author Response
Comment 1:
Main weakness: The presentation of Results Sections 2.1 and 2.2 is long-winded and difficult for the reader to follow, undermining the main subject of this manuscript.
Suggestion: Rewrite it and combine them together. Move the figure 1, figure 2 and table 1 to supplemental data. Keep the figure 3, figure 4 and make everything simple. Point the most important finding is enough.
Response:
Dear Reviewer, many thanks indeed for your time and suggestions. We have deleted some text from the results section that can be explained by the table 1, in order to avoid repetition and ease the reading as suggested. Figure 2 is now supplementary Figure 4, and sections 2.1 and 2.2 have been merged. The minor errors pointed out have also been ironed out. We hope that you may find this satisfactory.
Comment 2: By the way, the iThenticate similarity of this manuscript is 75% that is super high. Is this manuscript published in another place?
Response:
- Dear Reviewer, we would like to note that the similarity is because a preprint version of this manuscript was published on Research Square (https://www.researchsquare.com/article/rs-3894515/v1)
Round 2
Reviewer 1 Report
Comments and Suggestions for Authors
The authors have addressed my comments
Author Response
Thank you very much for your valuable comments and suggestions. We appreciate your time and efforts.
Reviewer 3 Report
Comments and Suggestions for Authors
This manuscript aims to elucidate the changes in transcriptomic milieu and cytokine profile induced by Vpx through cell lines. Vpx is a virus-associated accessory protein that is essential for infection and play a crucial role in regulating the immune response of host cells.
This manuscript found Vpx differentially altered several genes expression that involved in regulation of viral process and innate immune response of host. This study provides important information about the complex role played by Vpx in priming and taming the cellular environment to establish infection. The authors used transcriptome analysis to address Vpx overexpressing THP-1 cells, making the subject original. The comparison of K68-R70 function-restricted protein with wild-type protein makes it scientific solid. These all relevant to the field and the references are appropriate, too.
Over all, it is a good manuscript.
Author Response
We appreciate your comments and would like to thank you for your time and efforts.